# Frequency-Aware Token Reduction for Efficient Vision Transformer

**Dong-Jae Lee**[1]     **Jiwan Hur**[1]     **Jaehyun Choi**[1]     **Jaemyung Yu**[2]     **Junmo Kim**[1]

[1]KAIST     [2]NAVER AI Lab

{jhtwosun, jiwan.hur, chlwogus}@kaist.ac.kr,
jaemyung.yu@navercorp.com, junmo.kim@kaist.ac.kr

## Abstract

Vision Transformers have demonstrated exceptional performance across various computer vision tasks, yet their quadratic computational complexity concerning token length remains a significant challenge. To address this, token reduction methods have been widely explored. However, existing approaches often overlook the frequency characteristics of self-attention, such as rank collapsing and over-smoothing phenomenon. In this paper, we propose a frequency-aware token reduction strategy that improves computational efficiency while preserving performance by mitigating rank collapsing. Our method partitions tokens into high-frequency tokens and low-frequency tokens. high-frequency tokens are selectively preserved, while low-frequency tokens are aggregated into a compact direct current token to retain essential low-frequency components. Through extensive experiments and analysis, we demonstrate that our approach significantly improves accuracy while reducing computational overhead and mitigating rank collapsing and over smoothing. Furthermore, we analyze the previous methods, shedding light on their implicit frequency characteristics and limitations. The code is available in https://github.com/jhtwosun/frequency-aware-token-pruning.

## 1 Introduction

The advent of Vision Transformers (ViTs) [8] has marked a significant milestone in computer vision, demonstrating remarkable performance on various benchmarks. Following this success, research has focused on utilizing ViTs in various applications. However, their computational overhead has significantly hindered their deployment in real-world applications, primarily due to the quadratic complexity of self-attention with respect to token length. This has motivated extensive work on token-reduction, which aims to lower computational cost by discarding or aggregating tokens with minimal impact on performance. Prior token-reduction methods fall into two main categories: token merging and pruning. Merging methods fuse similar or neighboring tokens, whereas pruning methods remove less informative tokens and retain only the most important ones. Both strategies have demonstrated significant computational savings with minimal performance loss.

Meanwhile, another line of research has focused on the theoretical properties of ViTs to analyze their characteristics. Recent studies have identified a *rank-collapsing* phenomenon, in which the self-attention layer in a Transformer causes the feature to converge to a rank-1 matrix, where all tokens share the same representation [6]. From a frequency-domain perspective, researchers [26, 18] showed that stacking self-attention blocks is analogous to applying a repeated low-pass filter. Consequently, self-attention primarily preserves the direct current (DC) component, i.e., the zero-frequency component of the input, leading to an over-smoothing of feature map. This leads to a loss of token-level information, limiting the model's capacity and expressive power. Subsequent work has demonstrated that mitigating rank collapse and integrating both low- and high-frequency components can significantly improve the performance of ViT [34, 19, 32, 18, 26].

39th Conference on Neural Information Processing Systems (NeurIPS 2025).

Considering such analysis, token reduction should avoid discarding tokens containing relatively high-frequency information, as this can expedite over-smoothing and limit expressive power. However, previous token reduction methods overlook such connection with high-frequency information and rank collapse. Consequently, they fail to preserve such information effectively, accelerating the rank collapse. Merging-based approaches naturally suppress high-frequency signals during token aggregation, accelerating rank-collapsing. Pruning-based methods may retain high-frequency information if the preserved tokens capture high-frequency content. However, as we later show (Section 6.3), we observe that pruning methods preserve high-frequency tokens only in the near-last layers, and their behavior varies substantially across different layer depths.

In this paper, we propose a frequency-aware token reduction, which focuses on mitigating the over-smoothing phenomenon and preserving the expressive power of ViTs while lowering their computational cost. Specifically, we partition the token set into two subsets: one containing primarily high-frequency information and another dominated by low-frequency content. By selectively retaining high-frequency tokens and reducing low-frequency tokens, we minimize the loss of high-frequency signals, thereby alleviating rank collapse and over-smoothing phenomenon. To further mitigate the information loss from discarding low-frequency tokens, we aggregate their DC signals into a single token—acting as a compact representation of the low-frequency information.

Through extensive analyses and experiments across various models, we demonstrate that the proposed method effectively alleviates rank collapse and preserves accuracy while reducing computational costs. Furthermore, we analyze previous methods from a frequency perspective, shedding light on their frequency characteristics and limitations.

## 2 Background

### 2.1 Vision Transformer

A Vision Transformer (ViT) block consists of two primary components: Multi-Head Self-Attention (MSA) and a Feed-Forward Network (FFN). Typically, the ViT transforms an input image $I \in \mathbb{R}^{H \times W \times C}$ into tokens $X \in \mathbb{R}^{n \times d}$ through patch embedding, $n = HW/p^2$ represents the number of tokens where $p$ is the patch size, and $d$ is the feature dimension. Special tokens (e.g., the CLS token) may optionally be appended to facilitate tasks such as classification.

The key characteristic of the transformer lies in its Self-Attention (SA) layer. SA encodes the token representations from the previous layer $X \in \mathbb{R}^{n \times d}$ by aggregating information from other tokens via an attention weight matrix $A \in \mathbb{R}^{n \times n}$. Formally, it is defined as [25]:

$$\text{SA}(X) = AXW_V \tag{1}$$

$$= \text{softmax}\left( \frac{XW_Q(XW_K)^T}{\sqrt{d_k}} \right) XW_V, \tag{2}$$

where $W_Q \in \mathbb{R}^{d \times d_q}$, $W_K \in \mathbb{R}^{d \times d_k}$, and $W_V \in \mathbb{R}^{d \times d_v}$ are the query, key, and value weight matrices, respectively, and $d_q, d_k, d_v$ denote the dimensions of the query, key, and value vectors, respectively. The term $\sqrt{d_k}$ serves as a scaling factor, and softmax$(\cdot)$ is applied row-wise. MSA aggregates outputs from multiple SA heads and projects them into the hidden dimension via a linear transformation.

### 2.2 Over-smoothing of self-attention

Extensive studies show that stacking SA leads to rank collapse or over-smoothing phenomena, which reduces the token diversity and hinders the model from learning rich features [6]. Formally, it is shown that the following proposition holds [7, 26].

**Proposition 2.1.** *Let the mean-centered matrix of the feature matrix $X$ be $H_f[X] = (I - \frac{1}{n}\mathbf{1}\mathbf{1}^T)X$, which can also be viewed as a high-pass filtered version of $X$. Then, SA reduces the high-frequency component with collapsing ratio $\lambda$ by:*

$$\|H_f[\text{SA}(X)]\|_F \leq \lambda \|H_f[X]\|_F, \tag{3}$$

For ViTs, researchers [26, 18] showed that the attention matrix $A$ behaves as a low-pass filter, preserving the direct component (DC) signal, i.e., the zero-frequency component at the output layer,

causing an over-smoothing problem. As the collapsing phenomenon becomes more severe, $\boldsymbol{A}$ becomes closer to a low-pass filter, i.e., $\boldsymbol{A} = \frac{1}{n}\mathbf{1}\mathbf{1}^{\top}$. Consequently, SA fails to capture meaningful token interactions, causing the output features to become similar across adjacent layers. A deeper background is available in the Appendix A.

To address this issue, recent research has emphasized the importance of counteracting over-smoothing by balancing low-frequency and high-frequency components, which can significantly enhance the performance of ViTs. Various approaches have been proposed, including designing modules to emphasize high-frequency information, decomposing $\boldsymbol{A}$ into low-pass and high-pass filters, incorporating convolution layers, and improving training strategies [18, 26, 32, 19, 34].

# 3 Frequency-Aware Token Reduction

## 3.1 Token reduction in frequency aspect

Figure 1 provides an empirical visualization of the rank-collapsing phenomenon. The amplitude of high-frequency components decreases with increasing layer depth. A similar trend is observed when comparing the relative amplitude of high-frequency signals across different layers (Figure 1b). The slower decrease of high-frequency signals observed in the early layers can be attributed to their convolution-like characteristics. Specifically, early self-attention layers predominantly interact with spatially adjacent tokens, due to the characteristics of natural image [18]. As the layer goes deeper, the feature loses high-frequency information, leading to limitations in the token diversity.

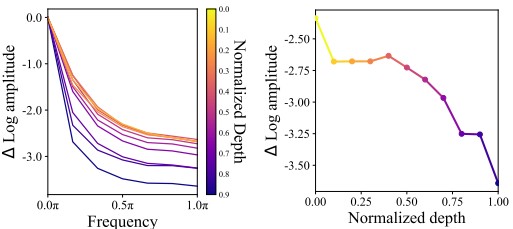

(a) $\Delta$ log amplitude     (b) Relative high-frequency

Figure 1: Frequency analysis of sthe elf-attention layer according to the layer depth. (a) Relative log amplitude of frequency. (b) Relative log amplitude of high-frequency $(1.0\pi)$.

Such rank collapsing and over-smoothing limit the model's capacity and expressive power. Therefore, preserving high-frequency signals is crucial for increasing the performance of ViTs [34, 19]. However, token reduction inherently leads to the loss of high-frequency signals, exacerbating rank collapse and diminishing token diversity. Existing token reduction methods overlook these frequency perspectives. Specifically, pruning and merging operations decrease high-frequency information unless the selected token $x_i$ is already collapsed, i.e., $x_i = \frac{1}{n}\sum_j x_j$, which leads to accelerating the collapsing. Formally, we have the following propositions.

**Proposition 3.1.** *Let $\boldsymbol{M} \in \mathbb{R}^{n' \times n}$ denotes row-normalized matrix, where $n' < n$. If the $(i,j)$ element of $\boldsymbol{M}$ is binary, i.e. $m_{i,j} \in \{0,1\}$, $\boldsymbol{M}$ serve as a row-selection matrix, and $\boldsymbol{M}\boldsymbol{X}$ represents the token pruning. Similarly, if $m_{i,j} \in (0,1)$, then $\boldsymbol{M}\boldsymbol{X}$ represents the token-merging. Then, we have:*

$$\|H_f[\mathrm{SA}(\boldsymbol{M}\boldsymbol{X})]\|_F \leq \|H_f[\mathrm{SA}(\boldsymbol{X})]\|_F \leq \lambda \|H_f[\boldsymbol{X}]\|_F. \tag{4}$$

Derivation of Equation (4) can be found in Appendix B. Intuitively, merging-based methods reduce token diversity thereby accelerating rank collapse. Pruning-based methods are similarly susceptible if the tokens removed predominantly contain high-frequency signals. Consequently, token reduction accelerates the rank collapsing, hindering the model's ability to represent diverse features and limiting the model's capacity and expressive power.

Given these limitations, a more effective approach should explicitly consider the frequency aspect when performing token reduction. To address these issues, we propose a novel approach that explicitly selects and retains tokens contributing predominantly to high-frequency components while merging those associated with low-frequency information. This frequency-aware token reduction strategy aims to mitigate rank collapse while reducing computational costs.

## 3.2 Selection of Frequency token

For frequency-aware token-reducing, we first separate tokens into two sets: those primarily containing high-frequency signals (HF tokens) and those dominated by low-frequency signals (LF tokens). For separating the token sets, a naïve method involves calculating the DC signal of the entire feature and

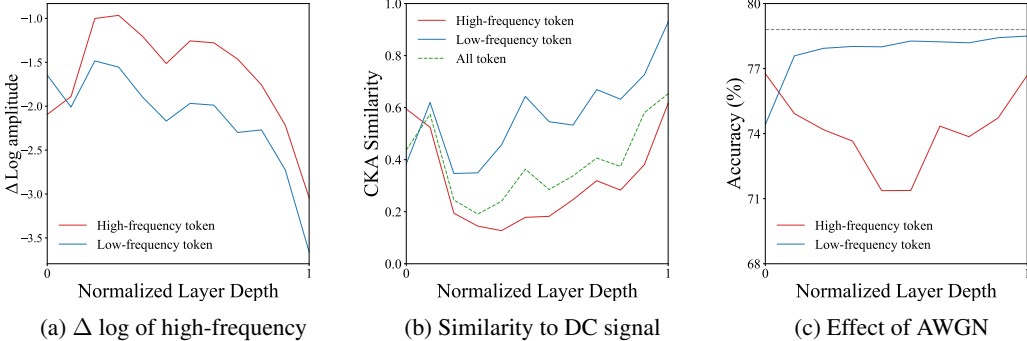

(a) Δ log of high-frequency     (b) Similarity to DC signal     (c) Effect of AWGN

Figure 2: Analysis of HF tokens and LF tokens. (a) Relative log amplitude of high-frequency $(1.0\pi)$. HF tokens contain relatively more high-frequency signals than LF tokens. (b) Similarity with DC-signal. The LF tokens dominantly contain the DC signal of features compared to the HF tokens. (c) Effect of white Gaussian noise (AWGN) on each token set. We report a mean accuracy of 10 trials; the gray dashed line represents the initial accuracy. The results show that the HF tokens are more critical in maintaining the model accuracy compared to the LF tokens.

then measuring the similarity of each token to it. Tokens exhibiting high similarity can be classified as LF tokens. However, this method introduces additional computational overhead, which contradicts the primary objective of reducing computational cost in token reduction.

Instead, we decompose the attention map $\boldsymbol{A}$ into a low-frequency attention map $\boldsymbol{A}^{LP}$ and a high-frequency attention map $\boldsymbol{A}^{HP}$:

$$\boldsymbol{A}^{LP} = \frac{1}{n}\boldsymbol{1}\boldsymbol{1}^T, \tag{5}$$

$$\boldsymbol{A}^{HP} = \boldsymbol{A} - \boldsymbol{A}^{LP}. \tag{6}$$

In the frequency view, $\boldsymbol{A}^{LP}\boldsymbol{X}$ represents the zero-frequency component (DC bias) of $\mathcal{F}(\boldsymbol{AX})$, where $\mathcal{F}$ represents discrete fourier transforms. The residual term, $\boldsymbol{A}^{HP}\boldsymbol{X}$, represents the residual high-frequency terms. In the matrix $\boldsymbol{A}^{HP} \in \mathbb{R}^{n \times n}$, the $(q, k)$-th element quantifies the contribution of the $k$-th token to the high-frequency component of the $q$-th output token. Therefore, if a certain column of $A_{HP}$ has relatively larger weights than other columns, the token corresponding to this column contributes more to the high-frequency components of the output tokens and can be classified as an HF token. Conversely, tokens with lower weights are classified as LF tokens.

We select the $r$ tokens with the highest weights in $\boldsymbol{A}^{HP}$ as the HF token set, denoted as $N_{HF}$, and the $r$ tokens with the lowest weights as the LF token set, denoted as $N_{LF}$. Formally, the selection process can be described as:

$$\tilde{A}_k = \frac{1}{nh}\sum_{h'=1}^{h}\sum_{n'=1}^{n} A_{n',k}^{HP(h')} \tag{7}$$

$$N_{HF} = \operatorname*{argmax}_{S \subseteq \{1,\ldots,n\}, |S|=r} \sum_{k \in S} \tilde{A}_k, \tag{8}$$

$$N_{LF} = \operatorname*{argmin}_{S \subseteq \{1,\ldots,n\}, |S|=r} \sum_{k \in S} \tilde{A}_k, \tag{9}$$

The summation of Equation (7) is applied column-wise (over query dimension) with respect to attention map of each heads $\boldsymbol{A}^{HP(h)}$, and $\tilde{A}_k$ captures importance of $k$th token in terms of contribution to high frequency component.

For more precise justification, refer to Appendix C. Empirically, we validate the high- and low-frequency token sets using ViT [8] and ImageNet-1K validation set [5]. We set $r = \lfloor n\tau \rfloor$ with $\tau = 0.25$. Fourier analysis reveals that tokens classified as LF tokens exhibit significantly lower high-frequency components than HF tokens (Figure 2a). While both HF and LF tokens show a decline in high-frequency components as layer depth increases—consistent with the rank collapse

phenomenon—HF tokens consistently retain more high-frequency signals than their LF counterparts. This confirms that HF tokens are more effective in preserving high-frequency features. The first layer has the opposite behavior, since the input to the first layer is an embedding of an image via convolution. Similar to convolution, the attention layer here shows stronger attention to spatially adjacent tokens [18]. Additionally, we measured the similarity of each token to the feature's DC signal (Figure 2b). The analysis revealed that LF tokens contain significantly more DC signal components than HF tokens, confirming that our method effectively separates tokens based on their frequency characteristics without introducing excessive computational overhead.

To assess the functional importance of these token sets, we evaluated the model's accuracy after applying additive White Gaussian Noise (AWGN) to each token set (Figure 2c). The results highlight the critical role of HF tokens in maintaining model accuracy. Since the selection method proposed in Equations (7) to (9) requires only simple averaging, it is much more efficient than those methods than rely on Fourier transform or cosine similarity while achieving the similar effect of selecting high frequency components as evidenced by the results in Figure 2.

### 3.3 Frequency-Aware Token Reduction

Building on our analysis, we propose a token reduction method that retains HF tokens while preserving the DC components of the remaining tokens through a single DC token. Specifically, we retain the HF token set, $N_{HF}$, as defined in Equation (8). However, removing LF tokens ($N \setminus N_{HF}$) disrupts the DC signals of the original features (Figure 2b). While these signals are less critical than high-frequency information, they still have a meaningful impact on accuracy (Figure 2c). To address this issue, we propose aggregating the DC signals of LF tokens into a single DC token. This approach compactly preserves essential DC information, ensuring that the overall feature representation remains balanced. By combining HF token preservation with DC signal aggregation, we can minimize the information loss while maintaining model accuracy and reducing computational costs.

In addition, some LF tokens may still contain high-frequency signals in the early layers, as shown in Figure 1. Thus, treating all LF tokens as purely low-frequency signals may lead to the unintended loss of critical high-frequency information. To address this issue, we leverage spatial locality—a characteristic of early layers in vision transformers [18, 20]—by introducing local DC tokens.

To achieve this, we divide the token set into $w^2$ spatially adjacent groups, where the $j$th local group is denoted as $N^j$. Each local group, comprises $N^j \subset \{1, \ldots, n\}$ is a set of token index for the $j$th group, which has $n/w^w$ tokens, i.e. $|N^j| = n/w^2 = H/wp \cdot W/wp$ tokens. The LF tokens in each local group, $N_{LF}^j = N_{LF} \cap N^j$, are aggregated into a local DC token as follows:

$$X_{DC} = \left\{ x_{DC}^j \middle| x_{DC}^j = \frac{1}{|N_{LF}^j|} \sum_{i \in N_{LF}^j} x_i, j \in \{1, \ldots, w^2\} \right\}. \tag{10}$$

When $w = 1$, this results in a single global DC token. By employing local DC tokens, we minimize high-frequency signal loss in early layers while still reducing the token count. The reduced token set $\tilde{N}$, comprising $|N_{HF}|$ HF tokens and $w^2$ DC tokens, is passed to the next layer where $|N_{HF}| = \lfloor n\tau \rceil$ with $\tau$ represents the reducing ratio.

In cases where token reduction is applied across multiple layers $(l-1, l)$, HF token selection process remains unchanged, while the DC tokens in the $l$th layer $X_{DC,l}$, are updated as follows:

$$X_{DC,l} = \left\{ x_{DC,l}^j \middle| x_{DC,l}^j = \frac{1}{|N_{LF,l}^j| + 1} \left( \sum_{i \in N_{LF,l}^j} x_{i,l} + x_{DC,l-1}^j \right), j \in \{1, \ldots, w^2\} \right\}, \tag{11}$$

This iterative update ensures that the information from LF tokens is retained in the subsequent layers.

We aim to apply the proposed method to pretrained models while mitigating the rank collapse problem. However, self-attention remains prone to collapsing the remaining HF tokens. To address this issue, we modify the attention weight matrix $\hat{A}$ as follows:

$$\hat{A} = A^{LP} + (\omega_1 + 1)A^{HP} + (\omega_2 + 1)A^{N_{DC}}, \tag{12}$$

where $\omega_1, \omega_2 \in \mathbb{R}^h$ are learnable parameters, and $A^{N_{DC}}$ represents the zero-padded attention weights assigned to the DC tokens. The parameter $\omega_1$ ensures that HF tokens are emphasized to prevent

collapse, while $\omega_2$ adjusts their attention weights when DC tokens are present from previous layers. This adjustment accounts for the fact that DC tokens, derived through aggregation, tend to receive lower attention scores due to Jensen's inequality—i.e., the attention weight assigned to the DC token is lower than the sum of the attention weight assigned to the individual LF token. Therefore, we introduce $\omega_2$ to allow the model to adjust low-frequency information appropriately.

# 4    Related Works

Token reduction methods aim to improve the efficiency of Transformers. They aim to reduce the quadratic complexity of self-attention to token length while minimizing accuracy degradation without modifying the architecture, making them directly applicable to various pretrained models.

A core idea of token reduction can be categorized into two: first, merging-based approaches, such as ToME [1], DTEM [13], and DiffRate [2] calculate the token similarity and merge them to reduce redundancy. The related approach involves spatial pooling, which also reduces token length but is not directly applicable to pretrained models since it significantly alters model behavior and learned features. As a result, spatial pooling is more commonly employed in designing efficient architectures rather than directly reducing token length in pretrained models [15].

The second category focuses on token pruning, where only the most informative tokens are retained while discarding redundant ones. DynamicViT [21] and AdaViT [16] introduce additional layers to select the informative token with a fixed pruning ratio. More recently, EViT [14], ATS [10], and Evo-ViT [29] have leveraged the CLS token as a selection marker, making their methods efficient with few additional costs. A-ViT [30] and ATS [10] further improve the previous methods with dynamic pruning. Several approaches have been proposed to mitigate the potential loss of information from discarded tokens. Evo-ViT, SPViT, EViT, and TPS [27] additionally preserve the discarded tokens by utilizing them in the residual connection or fusing them into remaining tokens.

# 5    Experiments

In this section, we validate the effectiveness of the proposed method in various models. For discussion and ablation study, we used the widely used configuration as the default configuration, else if otherwise noted. The results of the experiments with the searched Pareto-optimal configuration are available in Appendix F. For more experimental results regarding the throughput, additional ablation studies, and application to the downstream tasks, please refer to Appendices G to I.

## 5.1    Experimental Setup

To validate the effectiveness of our proposed method, we conduct experiments on ImageNet-1K [5] and compare our results with state-of-the-art methods. Specifically, we utilize pretrained models on ImageNet-1K and fine-tune them for 30 epochs. For initial experiments, we apply the common practice of applying reductions at the 4th, 7th, and 10th layers, reducing the token count by $30\%$ at each layer. We evaluate our approach across different model sizes using DeiT [24]. Additionally, we assess performance across different training strategies, including: DeiT, ViT[22], ViT pre-trained with ImageNet-21K and finetuned with -1K, and ViT trained with self-supervised learning (MAE, DINO) [11]. For other experimental details, we follow prior works [14, 29, 21]. Specifically, we employ self-distillation, where the original model serves as a teacher model. We set the local window size $w = 2, 1, 1$ for all experiments except ViT-S, which is used with $w = 2, 2, 1$. For quantitative evaluation, we report Top-1 accuracy (%), the number of Multiply-Accumulate operations in giga units (MACs), and the number of parameters in a million units. All experiments are conducted using training scripts and pretrained models from `timm` [28], and MACs are measured using `torchprofile`. Further details on the experimental procedures can be found in Appendix D.

## 5.2    Experimental Results

**Comparison with SOTA methods**    In Table 1, we compare with the recent token-reducing SOTA methods. The proposed method demonstrates better performance compared to existing approaches. In certain cases, it even outperforms the original model, which can be attributed to its ability to mitigate rank collapse effectively. Although the method introduces additional components, such as

Table 1: Comparison of various token reduction methods and the proposed method across models. For models trained with self-supervised models, we fine-tuned the baseline model with 30 epochs.

| Model | Method | Params | MACs | Acc. (%) |
|---|---|---|---|---|
| | Baseline | 5.6 | 1.3 | 72.2 |
| DeiT-T | Dy. ViT | 5.9 | 0.8 | 71.4 |
| | EVO-ViT | 5.9 | 0.8 | 72.0 |
| | ToMe | 5.6 | 0.8 | 71.4 |
| | EViT | 5.6 | 0.8 | 71.9 |
| | **Ours** | 5.6 | 0.8 | **72.3** |
| | Baseline | 22.1 | 4.6 | 79.8 |
| DeiT-S | Dy. ViT | 22.8 | 2.9 | 79.3 |
| | EViT | 22.1 | 3.0 | 79.5 |
| | ToMe | 22.1 | 2.9 | 79.5 |
| | ATS | 22.1 | 2.9 | 79.7 |
| | METR | 22.1 | 3.0 | 79.6 |
| | DTEM | 22.1 | 3.0 | 79.4 |
| | Zero-TP | 22.1 | 3.0 | 79.4 |
| | DiffRate | 22.1 | 2.9 | 79.6 |
| | Evo-ViT | 22.1 | 3.0 | 79.4 |
| | **Ours** | 22.1 | 3.0 | **79.8** |
| | Baseline | 86.6 | 17.6 | 81.8 |
| DeiT-B | Dy. ViT | 89.5 | 11.5 | 81.4 |
| | ToMe | 86.6 | 11.5 | 81.4 |
| | EViT | 86.6 | 11.6 | 81.4 |
| | DiffRate | 86.6 | 11.5 | 81.5 |
| | **Ours** | 86.6 | 11.6 | **81.8** |

| Model | Method | Params | MACs | Acc. (%) |
|---|---|---|---|---|
| | Baseline | 22.1 | 4.6 | 78.8 |
| ViT-S | ToMe | 22.1 | 3.0 | 77.7 |
| | EViT | 22.1 | 3.0 | 77.9 |
| | **Ours** | 22.1 | 3.0 | **79.0** |
| | Baseline | 22.1 | 4.6 | 81.1 |
| ViT-S-21K | ToMe | 22.1 | 3.0 | 80.1 |
| | EViT | 22.1 | 3.0 | 80.6 |
| | **Ours** | 22.1 | 3.0 | **81.2** |
| | Baseline | 86.6 | 17.6 | 83.7 |
| ViT-B-MAE | ToMe | 86.6 | 12.1 | 82.3 |
| | DiffRate | 86.6 | 12.1 | 82.9 |
| | **Ours** | 86.6 | 12.1 | **83.2** |
| | Baseline | 307 | 61.6 | 86.0 |
| ViT-L-MAE | ToMe | 307 | 42.3 | 85.6 |
| | DiffRate | 307 | 42.3 | 85.6 |
| | **Ours** | 307 | 42.3 | **85.7** |
| | Baseline | 86.6 | 17.6 | 81.9 |
| ViT-B-DINO | ToMe | 86.6 | 12.1 | 80.8 |
| | **Ours** | 86.6 | 12.1 | **81.4** |
| | Baseline | 26 | 6.6 | 83.3 |
| LV-ViT-S | EViT | 26 | 4.7 | 83.0 |
| | **Ours** | 26 | 4.7 | **83.3** |

local DC tokens and learnable parameters for attention weights, these additions are minimal and do not significantly increase the number of parameters or computational cost compared to existing methods. In the following sections, we analyze the proposed method in detail across various models.

**Comparison of different pretrained models**    Rank collapse is known to be influenced by various training hyperparameters [7, 26]. Therefore, we evaluate the proposed method across different pretrained models, including DeiT, ViT, and ViT pretrained on ImageNet-21K and MAE. We visualize the comparison of log amplitude of high-frequency with different models in Figure 3a. DeiT incorporates several techniques, such as drop path and strong data augmentation, which enhance performance and preserve patch diversity more effectively than ViT [3]. Applying the proposed method to these models reveals a more significant performance improvement in ViT than in DeiT, as ViT is more susceptible to rank collapse, amplifying the method's impact. Similarly, the ViT-21K model also exhibits better performance gains than DeiT. In contrast, for models pretrained via self-supervised methods, we observe varying impacts when applying the proposed method. This can be attributed to reduced rank collapse effects in these methods: MAE explicitly promotes token diversity through reconstruction tasks, and DINO encourages diverse yet consistent representations across augmented views. Nevertheless, our proposed frequency-aware approach minimizes frequency-related information loss compared to existing methods, resulting in relatively smaller performance drops. A detailed analysis of these pretrained models is provided in Appendix E.

**Comparison of different model sizes**    When comparing the performance of the proposed method across different sizes of DeiT, the proposed method improves the performance slightly despite reducing the computational cost. We also observed that smaller models exhibited better performance improvements. The differences between model sizes lie in the number of heads in MSA and channel size. Previous studies have shown that the rank collapsing rate is influenced by the Frobenius norm of the projection matrix in SA and the number of heads, with the latter particularly mitigating the collapsing rate [26]. Consequently, smaller models, which are more prone to rank collapse, show slightly better performance from the proposed method. We visualize the comparison of log amplitude of high-frequency with different model sizes in Figure 3b.

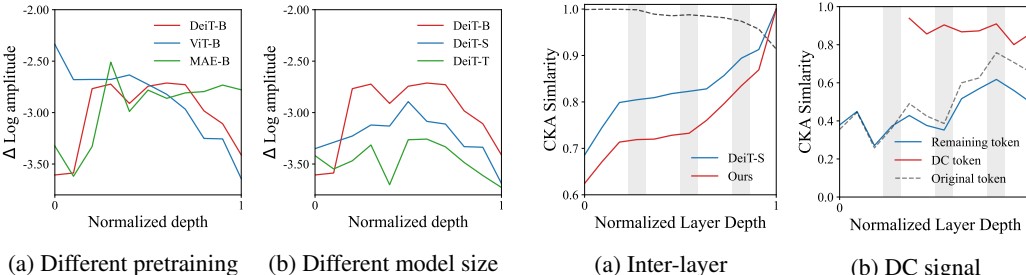

(a) Different pretraining     (b) Different model size     (a) Inter-layer     (b) DC signal

Figure 3: Relative log amplitude of high-frequency ($1.0\pi$) of models with (a) different training methods (ViT, DeiT, MAE) and (b) different model size (tiny, small and base).

Figure 4: (a) Similarity with last layer feature. The gray-dashed line represents the similarity between the models. (b) Similarity with the DC signal feature. In both figures, the gray square represents the reduction layer.

## 6 Discussion

### 6.1 Analysis on the rank collapsing

To assess the effectiveness of our method in mitigating rank collapse, we analyze feature similarity between the last and intermediate layers. Since rank collapse reduces token diversity, outputs from intermediate layers become increasingly similar to the last layer [34, 26]. As shown in Figure 4a, our method achieves lower similarity compared to DeiT-S, demonstrating reduced rank collapse. Moreover, at deeper layers, our approach progressively learns more distinct features, effectively alleviating redundancy seen in DeiT. Additionally, we measured similarity between DC signals from the original features and HF/DC tokens generated by our method (Figure 4b). HF tokens show lower similarity to the overall DC signal, confirming our method effectively selects HF content. Conversely, DC tokens closely resemble the DC signal, effectively preserving low-frequency information. These results confirm that our method mitigates rank collapse while reducing computational cost.

### 6.2 Ablation study

**Effect of high-frequency token**    To evaluate the impact of high-frequency (HF) and low-frequency (LF) components on model performance, we conducted an ablation study where only DC tokens were retained while HF tokens were removed (Figure 5a). The results clearly demonstrate that reducing HF tokens causes more significant performance degradation than reducing DC tokens, underscoring the critical role of high-frequency components in model performance. During fine-tuning, we observed that the accuracy of LF token preservation drops to nearly 20%. This suggests that ViTs inherently rely on high-frequency signals despite the rank collapsing. In contrast, methods that utilize spatial pooling [15, 31], which rely predominantly on low-frequency signals, learn distinct features and therefore require training from scratch.

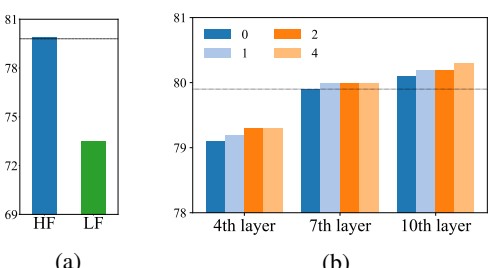

(a)          (b)

Figure 5: Results of the ablation study. The gray-dashed line represents the initial accuracy. (a) HF refers to the results of pruning the LF tokens, and LF refers to the results of pruning the HF tokens. (b) Effect of the number of the local window. Each number represents the window size.

**Effect of DC token**    We examined the importance of preserving DC signals by evaluating model performance without DC tokens (Figure 5b). We experiment with token reduction in a single layer with different local window sizes. The results confirm that DC tokens effectively retain the majority of the original DC signal, and their absence leads to notable performance degradation. Furthermore, we assessed the effect of local DC tokens under varying local window sizes. When reducing tokens in a single layer, performance improvements from local DC tokens were more pronounced in earlier layers compared to later layers. Considering the increased computational cost as the number of windows increases, we set $w$ as 2 in the earlier layer and 1 in other layers for the default configuration.

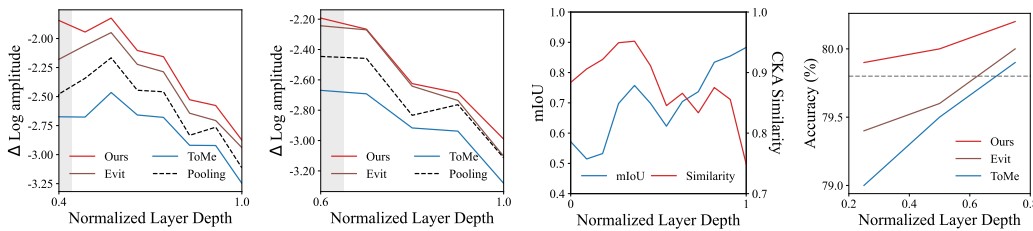

|                              |                              |                              |                              |
| :--------------------------: | :--------------------------: | :--------------------------: | :--------------------------: |
| (a) $4th$ layer Reduction    | (b) $7th$ layer Reduction    | (c) Comparison with CLS      | (d) Single-layer pruning     |

Figure 6: Analysis of the proposed method and the previous methods. (a),(b): Relative log amplitude of high-frequency in single-layer token reduction in $4th$ and $7th$ layer, respectively. (c): Selection mIoU with EViT and similarity of DC signal and CLS token. (d) Accuracy comparison between different methods with the single-layer token reduction in $4th$, $7th$, and $10th$ layers. The gray-dashed line represent the initial accuracy of DeiT-S.

## 6.3 Analysis on previous method

We conducted a detailed analysis of existing token reduction methods and compared them with the proposed approach. Specifically, we compare our method with ToMe [1], which merges similar tokens, and EViT [14], which identifies tokens with lower CLS attention weights as less important and removes them. For comparison, we fine-tuned all models with 30 epochs. To analyze frequency components, we also compare our approach with adding a spatial pooling layer in a token-reducing layer in DeiT. For analysis, we perform single-layer token reduction at different depths with a fixed reduction ratio of 50%. For the local window size, we set $w = 1$ for all layers.

When comparing the relative amplitude of high-frequency components (Figures 6a and 6b), the merging-based method exhibits lower high-frequency components than both the token selection-based approach and the proposed method. Furthermore, ToMe demonstrates even lower high-frequency amplitude than the model with pooling, which accelerates rank collapse and significantly impacts accuracy (Figure 6d). Applying merging strategies in shallower layers leads to early-layer rank collapse, causing a steeper decline in accuracy as the reduction layer moves to earlier layers.

On the other hand, EViT exhibits a high-frequency amplitude similar to that of the proposed method. EViT retains tokens that achieve high attention weights from the CLS token, assuming that these tokens are informative. To further investigate the relationship between the proposed method and EViT, we measured i) the intersection-over-union (IoU) between tokens selected by EViT and HF tokens selected by our method, and ii) the similarity of the DC signal with the CLS token. As shown in Figure 6c, the token sets differ in early layers but become increasingly similar at deeper layers. Our analysis reveals that the CLS token gradually incorporates high-frequency signals as depth increases. As a result, the similarity between the CLS token and the DC signal decreases with depth, leading to greater overlap in token selection. This explains why accuracy differences become less significant when tokens are reduced at deeper layers compared to earlier layers (Figure 6d).

## 6.4 Limitation

While our frequency-based approach successfully reduces computational costs and minimizes performance degradation, our experiments were limited to vision models. Since rank collapse is inherent to all transformer-based models, the proposed method could be applied to other domains, such as multi-modal models. Future work will focus on exploring adaptations to different modalities.

## 7 Conclusion

In this paper, we proposed a frequency-aware token reduction method that effectively reduces the computational cost of ViTs while mitigating the rank collapsing and over-smoothing phenomenon. Our approach explicitly selects and retains the HF token set and utilizes the DC token as a compact representation of low-frequency signal, along with a learnable attention mechanism. Experimental results demonstrated consistent improvements across various model sizes and pretrained configurations, effectively mitigating the rank-collapse while reducing the computational cost.

## Acknowledgements

This research was partly supported by Artificial intelligence industrial convergence cluster development project funded by the Ministry of Science and ICT (MSIT, Korea) & Gwangju Metropolitan City, and by the Institute of Information & Communications Technology Planning & Evaluation (IITP) grant funded by the Korea government (MSIT) (No.RS-2025-02283048, Developing the Next-Generation General AI with Reliability, Ethics, and Adaptability)

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

# A More Background

In this section, we delve deeper into the rank-collapsing phenomenon observed in Transformer architectures and discuss the mechanisms that mitigate this issue. As outlined in Section 2, Transformers can experience rank collapse, where token representations become nearly identical as the network depth increases. This reduction in representation diversity impairs the model's ability to capture complex features. Formally, we have following theorem.

**Theorem A.1.** *(Rank Collapse in Pure Attention Networks) Let $X^{(\ell)} \in \mathbb{R}^{n \times d}$ denote the token embedding matrix at layer $\ell$ (with $n$ tokens and embedding dimension $d$) and $H_c[X]$ the high-frequency component of the token feature matrix $X$. Then, self-attention operation ($SA$) reduces this component by:*

$$\|H_c[SA(X)]\|_F \leq \lambda \|H_c[X]\|_F, \tag{13}$$

*where the convergence rate $\lambda < 1$ occurs at a doubly-exponential rate in terms of the number of layers $\ell$ [6, 26].*

Dong et al. [6] employed a path-decomposition approach, showing that in pure attention networks, all but one component in the embedding space decay doubly exponentially. The dominant surviving component corresponds to uniform embeddings across tokens, resulting in rank collapse. Complementarily, Wang et al. [26] provided a frequency-domain analysis, demonstrating that repeated self-attention layers inherently function as low-pass filters, exponentially attenuating high-frequency differences between token embeddings. This further explains the rapid convergence toward uniform embeddings.

In addition, Noci et al. [17] found that rank collapse hinders training by causing the gradients of the queries and keys to vanish at initialization, making it difficult for the model to learn meaningful attention patterns. They suggest that an appropriate depth-dependent scaling of the residual branches can prevent rank collapse and stabilize training. Dong et al. [6] demonstrated that models composed solely of self-attention (SA) layers tend to produce outputs that collapse to a rank-1 matrix, with all token representations converging to the same vector. This collapse occurs doubly exponentially with depth, severely limiting the model's expressiveness. Residual connections and feed-forward networks (FFNs) are integral components of the Transformer architecture that help prevent rank collapse. Residual connections preserve the output of previous layers, counteracting the tendency toward rank collapse by maintaining the diversity of token representations across layers. This preservation ensures that the model retains the capacity to learn complex patterns. FFNs contribute by increasing the Lipschitz constant of the transformation applied to token representations, slowing down the convergence to a rank-1 matrix and allowing the model to maintain higher-rank representations over greater depths.

For more theoretical analysis, please refer to the following papers [6, 26].

# B Proof of Equation (4)

Equation (4) claims that

$$\|H_f[SA(\boldsymbol{MX})]\|_F \leq \|H_f[SA(\boldsymbol{X})]\|_F \leq \lambda \|H_f[\boldsymbol{X}]\|_F. \tag{14}$$

More precisely, previous works on attention collapsing shows that $\|H_f[SA(\boldsymbol{X})]\|_F \leq \lambda \|H_f[\boldsymbol{X}]\|_F$. Equation (4) shows that token pruning and merging may accelerate such collapsing. In below, we provide the detailed proof of Equation (4).

Let $X \in \mathbb{R}^{n \times D}$ be a matrix of token embeddings, and let $X' = MX$ where $M \in \mathbb{R}^{n' \times n}$ is a row-normalized matrix which represents either a row-selection matrix (with elements in $\{0, 1\}$) or a row-normalized matrix (with elements in $[0, 1]$) with $n' < n$. Define $H_c[X]$ as the mean-centered (high-frequency component of $X$. Then, we have:

$$\|H_c[SA(X')]\|_F \leq \|H_c[SA(X)]\|_F. \tag{15}$$

*Proof.* Let $\mu = \frac{1}{n} \sum_{i=1}^{n} X_{i,:}$ and $\mu' = \frac{1}{n'} \sum_{j=1}^{n'} X'_{j,:}$ be the mean vectors of $X$ and $X'$, respectively. Then,

$$H_c[X] = X - \mathbf{1}_n \mu, \quad H_c[X'] = X' - \mathbf{1}_{n'} \mu'.$$

and by the definition,

$$\|H_c[X]\|_F^2 = \sum_{i=1}^n \|X_{i,:} - \mu\|^2, \quad \|H_c[X']\|_F^2 = \sum_{j=1}^{n'} \|X'_{j,:} - \mu'\|^2.$$

*Case 1: $M$ is a row-selection matrix.*

Then $X'$ is simply a subset of the rows of $X$. A standard fact from statistics is that the sample variance of any subset of points is at most the variance of the full set. Concretely, let $S \subseteq \{1, \ldots, n\}$ be the indices selected by $M$. Then

$$\mu' = \frac{1}{n'} \sum_{i \in S} X_{i,:},$$

and

$$\sum_{i \in S} \|X_{i,:} - \mu'\|^2 \leq \sum_{i \in S} \|X_{i,:} - \mu\|^2 \leq \sum_{i=1}^n \|X_{i,:} - \mu\|^2.$$

*Case 2: $M$ is row-normalized.*

Here each row of $M$ forms a probability distribution over $\{1, \ldots, n\}$, so each output row $X'_{j,:}$ is a convex combination of the original rows:

$$X'_{j,:} = \sum_{i=1}^n M_{j,i} X_{i,:}, \qquad \sum_{i=1}^n M_{j,i} = 1.$$

By the convexity of squared Euclidean norm,

$$\|X'_{j,:} - \mu'\|^2 = \left\|\sum_i M_{j,i}(X_{i,:} - \mu')\right\|^2 \leq \sum_i M_{j,i} \|X_{i,:} - \mu'\|^2.$$

Summing over $j = 1 \ldots n'$,

$$\sum_{j=1}^{n'} \|X'_{j,:} - \mu'\|^2 \leq \sum_{i=1}^n \|X_{i,:} - \mu'\|^2 \leq \sum_{i=1}^n \|X_{i,:} - \mu\|^2,$$

where the last inequality again uses that re-centering about the true mean $\mu$ minimizes the total squared deviation.

In both cases, the inequality $\|H_c[X']\|_F \leq \|H_c[X]\|_F$ holds. It then follows from Proposition 2.1 that $\|H_c[SA(X')]\|_F \leq \|H_c[SA(X)]\|_F$, as claimed.

$\square$

## C  Decomposing high- and low-frequency tokens

In this section, we provide a formal justification for decomposing the attention matrix $\boldsymbol{A}$ into a low-frequency component $\boldsymbol{A}_{LP}$ and a high-frequency component $\boldsymbol{A}_{HP}$, as defined below:

$$\boldsymbol{A}_{LP} = \frac{1}{n}\mathbf{1}\mathbf{1}^T, \tag{16}$$

$$\boldsymbol{A}_{HP} = \boldsymbol{A} - \boldsymbol{A}_{LP}. \tag{17}$$

Formally, let $\boldsymbol{A} \in \mathbb{R}^{n \times n}$ be an attention matrix derived from self-attention (SA), and let $\mathbf{1}$ be an $n$-dimensional vector of all ones. Define the matrices:

$$\boldsymbol{A}_{LP} = \frac{1}{n}\mathbf{1}\mathbf{1}^T, \quad \boldsymbol{A}_{HP} = \boldsymbol{A} - \boldsymbol{A}_{LP}.$$

Then, $\boldsymbol{A}_{LP}$ acts as a low-pass filter and $\boldsymbol{A}_{HP}$ acts as a high-pass filter in the token embedding space. We consider token embeddings $X \in \mathbb{R}^{n \times d}$, and apply the attention mechanism as:

$$X' = \boldsymbol{A}X. \tag{18}$$

First, note that $\boldsymbol{A}_{LP}X = \frac{1}{n}\mathbf{1}\mathbf{1}^T X$ computes the mean embedding across all tokens and broadcasts this mean to all tokens. Explicitly,

$$(\boldsymbol{A}_{LP}X)_{i,:} = \frac{1}{n}\sum_{j=1}^{n} X_{j,:}, \quad \forall i \in \{1,\ldots,n\}. \tag{19}$$

This operation corresponds exactly to extracting and replicating the Direct-Current (DC) or mean component, thus representing a low-pass filtering operation.

Consequently, the residual term:

$$\boldsymbol{A}_{HP}X = (\boldsymbol{A} - \boldsymbol{A}_{LP})X = \boldsymbol{A}X - \frac{1}{n}\mathbf{1}\mathbf{1}^T X, \tag{20}$$

subtracts the mean component from the attention result, leaving only the differences between tokens (high-frequency variations) and removing the DC component. Thus, by definition, $\boldsymbol{A}_{HP}$ acts as a high-pass filter.

Formally, the eigen-decomposition perspective further clarifies this intuition: $\boldsymbol{A}_{LP}$ has a single nonzero eigenvalue corresponding to the eigenvector $\mathbf{1}$, representing the DC component. The orthogonal complement of $\mathbf{1}$ corresponds to higher-frequency components, precisely captured by $\boldsymbol{A}_{HP}$.

Therefore, $\boldsymbol{A}_{LP}$ and $\boldsymbol{A}_{HP}$ indeed represent low-pass and high-pass filtering operations, respectively, as claimed.

## D  Experiments Details

We conduct the experiments with AdamW with learning rate $0.0001$ and weight decay $2 \times 10^{-5}$. The batch size is set to 128 per GPU. For other hyperparameters and data augmentations, we follow DeiT, except warmup epochs and stochastic depth, which we set as 0. For EMA, we set the decay factor to 0.9998. Additionally, we utilize the EMA model as distillation. All experiments are conducted in 8 NVIDIA RTX 4090 with mixed precision. We use standard cross-entropy and self-distillation for loss term from an unpruned model, following [14, 21, 27]. For DC tokens, we additionally utilize positional embedding terms. For extra distillation from the large teacher model, we follow [12]. Specifically, we change the self-distillation term to distillation loss term [24] from the teacher model.

In the experiments, we utilize pretrained ViT and DeiT. The ViT and DeiT share the same architecture with different data augmentations and regularizers. We provide some details of the differences, based on their papers. Note that we change the augmentations when finetuning them following their recipes, except the stochastic depth which we set as 0 following previous works.

Table 2: Hyper-parameters of ViT-S and DeiT-S

| Methods | ViT-S | DeiT-S |
|---|---|---|
| Optimizer | AdamW | AdamW |
| learning rate | 0.003 | 0.001 |
| Weight decay | 0.03 | 0.05 |
| Stoch. Depth | | 0.1 |
| Repeated Aug | | ✓ |
| Rand Augment | 2 / 0 | 9/0.5 |
| Mixup prob. | | 0.8 |
| Cutmix prob. | | 1.0 |
| Erasing prob. | | 0.25 |

# E    Frequency Analysis of models

As described in Section 5.2, rank collapsing, and over-smoothing phenomenons are affected by model size, training hyperparameters and strategies. Here, we report frequency analysis of different pretraine models and different models. For comparison of training strategies, we conduct the analysis with the base model. For comparsion of model sizes, we conduct the analysis with DeiT. As can be seen in the figure below, ViT suffers more from these phenomena than DeiT. On the other hand, MAE shows a lower decrement of high-frequency components compared to the supervised model (ViT, DeiT).

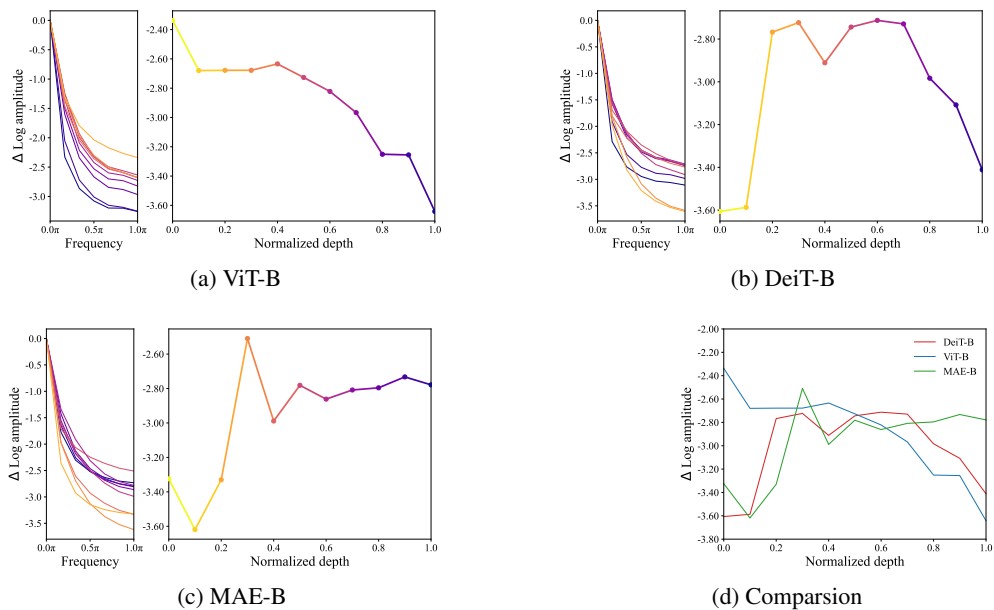

Figure 7: Frequency analysis of different pretrained model. (Left) Frequency analysis along layer depth. (Right) Relative high-frequency component along layer depth.

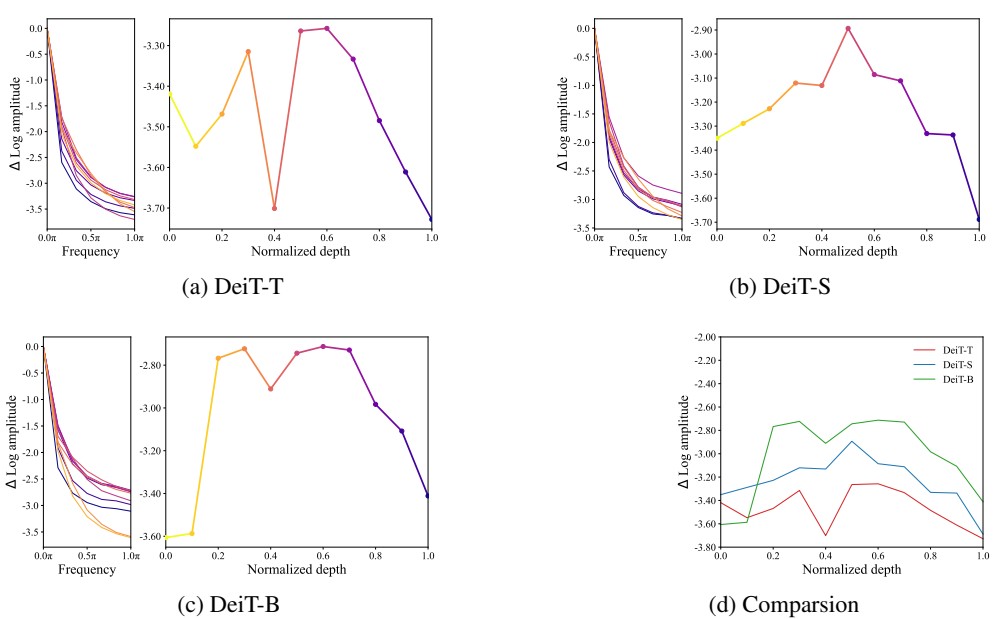

Figure 8: Frequency analysis of different model sizes. (Left) Frequency analysis along layer depth. (Right) Relative high-frequency component along layer depth.

## F   Results of the reducing configuration search

The proposed method can adjust the layer and ratio for token reduction. For initial experiments, we apply the common practice of applying reductions at the $4th, 7th$, and $10th$ layers, reducing the token count by $30\%$ at each layer. Additionally, we employed Neural Architecture Search (NAS) to identify Pareto-optimal reduction layers and ratios. The NAS optimization score was defined as:

$$\text{Score} = \left(1 - \frac{\text{MAC}_{current}}{\text{MAC}_{base}}\right) \times \frac{\text{Acc}_{current}}{\text{Acc}_{base}}. \tag{21}$$

The first term quantifies the relative reduction in computational cost using multiply-accumulate operations (MAC), while the second term evaluates the relative preservation of top-1 accuracy. To narrow the search space, we divided the layers into three groups ($2nd \sim 4th, 5th \sim 7th$ and $8th \sim 11th$) and searched for one reduction layer in each group with reducing ratio ($\{0.1, 0.2, \ldots, 0.9\}$) and local window size ($\{1, 2, 4\}$) to maximize the score.

We report the result of searched configurations for each DeiT model in the below table. We found that larger reduction ratios in deeper layers Here, we report the detailed results of one or two optimized configurations for each DeiT model. The results with "-o" is the results of the pareto-optimal configurations. The reducing ratio represents the token-reducing percentage of the *remaining tokens*. Our analysis reveals that employing larger reduction ratios in deeper layers leads to more Pareto-optimal outcomes. Notably, eliminating a greater number of low-frequency (LF) tokens at these depths incurs minimal information loss. This is attributed to the increased occurrence of rank collapse in deeper layers, which diminishes the presence of high-frequency (HF) tokens. Consequently, the higher incidence of rank collapse at these depths reduces the number of HF tokens, making the removal of additional LF tokens less detrimental. Interestingly, by expanding the size of the local window (i.e., increasing $|N_{DC}|$), we observed that even shallower layers could achieve optimal configurations with higher reduction ratios. This underscores the effectiveness of the proposed local DC token mechanism in preserving essential information while facilitating aggressive token reduction. Furthermore, our approach reduces the computational cost of DeiT-S to a level comparable with DeiT-T (ours-o2) while maintaining better accuracy.

Table 3: Results of the searched Pareto-optimal reducing configuration.

| Model | Method | Reducing layer | Reducing ratio | $w$ | GFLOPs | Acc. (%) |
|-------|--------|----------------|----------------|-----|--------|----------|
| DeiT-T | Baseline | N/A | N/A | N/A | 1.3 | 72.2 |
| | Ours | [4, 7, 10] | [30%, 30%, 30%] | [2, 1, 1] | 0.8 | 72.3 |
| | Ours-o1 | [4, 6, 9] | [50%, 40%, 20%] | [2, 2, 2] | 1.0 | 72.7 |
| | Ours-o2 | [4, 7, 9] | [50%, 40%, 40%] | [2, 2, 2] | 0.6 | 70.2 |
| DeiT-S | Baseline | N/A | N/A | N/A | 4.6 | 79.8 |
| | Ours | [4, 7, 10] | [30%, 30%, 30%] | [2, 1, 1] | 3.0 | 79.9 |
| | Ours-o1 | [4, 6, 10] | [40%, 50%, 90%] | [2, 1, 1] | 2.6 | 79.6 |
| | Ours-o2 | [2, 8, 10] | [70%, 60%, 60%] | [4, 2, 1] | 1.5 | 73.5 |
| DeiT-B | Baseline | N/A | N/A | N/A | 17.6 | 81.8 |
| | Ours | [4, 7, 10] | [30%, 30%, 30%] | [2, 1, 1] | 11.6 | 81.8 |
| | Ours-o1 | [4, 7, 11] | [60%, 50%, 50%] | [2, 2, 1] | 8.7 | 80.2 |

## G   Application to Dense Prediction

To assess the generality of our frequency-aware token reduction beyond image classification, we integrate it into a semantic segmentation pipeline. We choose the Segmenter framework [23] with an ADE20K [33]-pretrained ViT-B (EVA) encoder [9]. Our modifications occur during the Transformer encoding stage:

- **Token Pruning:** In each self-attention layer, we apply our frequency-aware pruning operator, removing a subset of tokens deemed low-value for high-frequency detail.

- **Low-Frequency Aggregation:** All pruned tokens' information is aggregated into a single token via averaging (the low-frequency component $\mathbf{A}_{LP}X$).

- **Residual Injection for Decoding:** To restore full spatial resolution for dense prediction, we inject the final token back into the decoder input as a residual. This ensures that low-frequency context flows into the upsampling and classification heads.

Table 4: Semantic segmentation results on ADE20K using Segmenter. Pruning configurations mirror common settings used in classification experiments.

| Model | Throughput (img/s) | mIoU (%) |
|---|---|---|
| Baseline (no pruning) | 21.7 | 51.3 |
| Config 1: 3% pruning on all layers | 68.0 | 51.1 |
| Config 2: 30% pruning on layers 4,7,10 | 92.1 | 50.8 |

These results show that our method can be effectively extended to dense prediction. By aggregating redundant tokens into a single low-frequency token, we remove unnecessary computation while preserving high-frequency details critical for segmentation. Even under 30% pruning at selective layers, the mIoU degrades by only 0.5 points, while throughput increases by over $4\times$. This demonstrates the practical utility of frequency-aware token reduction in segmentation and other downstream tasks without significant performance loss.

## H   Ablation on Attention Weight Modification

We conducted an ablation study on the DeiT-S model to examine the effects of attention weight modification. Our results indicate that the accuracy experienced a slight improvement from 79.8% (without reweighting ($\omega_1$) to 79.9% (with reweighting $\omega_1$). The primary objective of our proposed method is to mitigate the rank collapsing phenomenon by selectively preserving high-frequency tokens. The attention reweighting through $\omega_1$ provides additional support in alleviating this issue. Additionally, we observed that incorporating $\omega_2$ effectively reduced the number of epochs required for fine-tuning, with minimal impact on the final accuracy.

## I   End-to-End Throughput

Due to inconsistent code releases, some baseline methods report only GFLOPs; here we uniformly benchmark all methods under FP32 on an NVIDIA RTX 4090. The fastest result in each group is **bolded**, and the second-fastest is underlined.

Table 5: End-to-end throughput and accuracy for various token-reduction methods on ViT-S and ViT-B architectures (FP32 on RTX 4090).

| Method | imgs/s | Accuracy (%) |
|---|---|---|
| DeiT-S | 2803 | 79.8 |
| EViT | **3742** | 79.5 |
| ToMe | 3555 | 79.5 |
| DiffRate | 3538 | 79.6 |
| Ours | 3659 | **79.9** |

(a) ViT-S

| Method | imgs/s | Accuracy (%) |
|---|---|---|
| DeiT-B | 918 | 81.8 |
| EViT | **1338** | 81.4 |
| ToMe | 1223 | 81.4 |
| DiffRate | 1236 | 81.5 |
| Ours | 1310 | **81.8** |

(b) ViT-B

Our method consistently matches or exceeds both speed and accuracy of competing approaches, underscoring its practical deployability in real-world inference scenarios.

## J   Compatibility with FlashAttention [4]

Our frequency-aware attention modification integrates seamlessly with FlashAttention, requiring only a lightweight post-processing step without altering the core attention kernel. The proposed method rescales the attention map as follows:

$$\hat{A} = A_{LP} + (\omega_1 + 1) A_{HP} + (\omega_2 + 1) A_{NDC}. \tag{22}$$

In practice, after FlashAttention calculates the product $AV$, one needs only to calculate the mean value of each token set $N_{LF}$ and $N_{DC}$. Formally, we can compute the attention as follows:

$$\hat{A} = (\omega_1 + 1)\,A \; - \; \omega_1\,A_{LP} \; - \; (\omega_1 - \omega_2)\,A_{NDC}, \tag{23}$$

$$x = \hat{A}\,V = (\omega_1 + 1)\,A\,V \; - \; \omega_1 \sum_{i=1}^{n} V_i \; - \; (\omega_1 - \omega_2) \sum_{i \in N_{DC}} V_i. \tag{24}$$

This design preserves FlashAttention's low-memory, single-pass advantages and adds only two vectorized scaling operations.

For selecting the tokens $N_{HF}$ and $N_{LF}$, FlashAttention did not store the entire attention map, so we can directly apply our methods. However, we only need the $\texttt{argmax}$ in the column-wise value of the attention weight, not the exact value of the attention weight. Therefore, we only have to store the mean attention score $\tilde{A}$. Specifically, we store the block-mean attention score $S_j = \sum_i Q_i K_j^\top$. Alternatively, we can explicitly compute the $S_j = \sum_i Q_i K_j^\top$ out of the FlashAttention, for better generability with the existing frameworks. In either case, we can easily integrate the proposed methods with FlashAttention or any other modern attention-optimized techniques with minimal computational overhead. In the table below, we report the throughput of the proposed methods with FlashAttention and explicit computations. Since FlashAttention requires FP16, we changed the configurations of our experiments in the table below.

Table 6: Throughput (images/s) on an RTX 4090, FP16 with FlashAttention and our frequency-aware post-processing.

| Model | Configuration | Throughput |
|---|---|---|
| DeiT-S | w/ FlashAttention | 9303 |
| Ours | w/ FlashAttention | 12759 |

The results show that the proposed method is compatible with the modern attention-optimized methods.

## K  Visualization of HF/LF tokens

Following your valuable suggestion, we performed an experiment to visualize the selected High-Frequency (HF) and Low-Frequency (LF) tokens on sample images. While we cannot attach figures directly to this rebuttal, we will describe our key findings here and commit to adding these visualizations in the main manuscript. Our analysis confirmed that in early to mid-layers, the token selection aligns well with human intuition: HF tokens predominantly correspond to patches with object boundaries and complex textures, while LF tokens are clustered in smooth, uniform background areas. Interestingly, we also observed 'outliers' where some background patches were selected as HF tokens, which we found often correlates with high-norm features in the embedding space. In deeper layers, we observed that where tokens corresponding to the salient foreground of an object began to be classified as LF tokens. Since rank collapse becomes more severe, the unique signals of even the most important tokens in human perception, get 'smoothed out' and converge toward the low-frequency, DC-like representation of the entire feature map.

## L  Ablation on Different Image Size

To evaluate our method's effectiveness across different token lengths, we conducted an ablation study using various image and patch sizes. We utilized the ViT-21k-finetuned-on-1k model, which offers pre-trained weights for all tested configurations. The results, presented in Table 7, demonstrate that the benefits of our approach are most prominent with longer token sequences.

For instance, with a 16/384 patch/image size (576 tokens), our method improves accuracy from 86.6% to 87.0% while reducing computational cost by 35% (from 55.5 to 35.9 GFLOPS). This trend is even more pronounced for the 8/224 configuration (784 tokens), where we observe a 0.7% accuracy gain

with a 36% reduction in GFLOPS (from 78.3 to 50.0). The parenthesized values in the table represent GFLOPS.

Table 7: Performance comparison of ViT-B and ViT-S models with different patch and image sizes. Values are presented as accuracy (%) and a GFLOPS in parentheses.

| Patch Size / Image Size | 32/224 | 32/384 | 16/224 | 16/384 | 8/224 |
|---|---|---|---|---|---|
| Image Token Length | 49 | 144 | 196 | 576 | 784 |
| ViT-B (original) | 80.7 (4.4) | 83.4 (13.1) | 84.5 (17.6) | 86.6 (55.5) | 85.8 (78.3) |
| ViT-B (pruned) | 77.9 (3.2) | 84.0 (8.8) | 84.6 (12.1) | 87.0 (35.9) | 86.5 (50.0) |
| Difference | -2.8 (-1.2) | 0.63 (-4.3) | 0.14 (5.5) | 0.4 (-19.6) | 0.7 (-28.3) |
| ViT-S (original) | 76.0 (1.1) | 80.5 (3.4) | 81.1 (4.6) | 83.8 (15.5) | |
| ViT-S (pruned) | 73.5 (0.8) | 80.5 (2.3) | 81.4 (3.0) | 84.0 (9.9) | |
| Difference | -2.5 (-0.3) | 0.0 (-1.1) | 0.30 (-1.6) | 0.16 (-5.5) | |

