# OpenReview forum: "Frequency-Aware Token Reduction for Efficient Vision Transformer"
_NeurIPS.cc/2025/Conference — NeurIPS 2025 poster_

### Official Review · Reviewer_BU7E · 2025-06-19

**Clarity:** 2
**Significance:** 3
**Originality:** 4
**Rating:** 5
**Confidence:** 5

**Summary:**

This work examines the frequency behavior of self‑attention in Vision Transformers (ViTs), building on prior observations that self‑attention layers act like low‑pass filters—attenuating high‑frequency details and yielding progressively smoother outputs across layers. Noting that maintaining a balance between high‑ and low‑frequency information can boost performance, the authors investigate this balance in the context of token reduction. They quantify the relative importance of high‑frequency tokens versus low‑frequency (DC) tokens and introduce a streamlined method for distinguishing them, using the identity matrix as an efficient proxy for the DC component. By selectively pruning low‑frequency tokens while preserving high‑frequency ones, their frequency‑aware token reduction strategy outperforms conventional methods, both theoretically and empirically, underscoring the value of frequency‑based analysis for more effective token selection in transformer models.

**Questions:**

- *Token Selection with Windowing*: It remains unclear whether the high‑/low‑frequency distinction is applied globally before window partitioning or independently within each local window. Please specify the exact procedure—ideally with step‑by‑step pseudo‑code—to ensure reproducibility.
- *Role of DC Subtraction in Token Ranking*: In Section 3.2 (Lines 134–137), the authors use an argmax over each column of the high‑frequency attention map. Since argmax depends only on relative ordering, subtracting the low‑pass (all‑ones) component should not change the ranking. What, then, is the concrete benefit—computational or otherwise—of this subtraction? Please clarify or provide empirical evidence.
- *Precise Definition of “Relative Magnitude”*: In Section 3.2 (Line 140), the authors cite “relative magnitude” within columns of the high‑frequency attention map. Is this computed via column‑wise sums, means, norms, or another statistic? Please give a formal mathematical definition.
- *Visualization of Token Reduction*: It is strongly recommend adding qualitative visualizations—e.g., heatmaps showing which patches are deemed high‑ versus low‑frequency, before and after reduction. Such figures greatly enhance interpretability.
- *Computational Cost Analysis*: It is suggested to include a concrete analysis or empirical benchmark demonstrating the computational savings of your simplified frequency‑based token selection versus the naïve method (i.e., explicitly computing the DC component for every token).

**Ethical Concerns:**

["NO or VERY MINOR ethics concerns only"]

**Final Justification:**

The response addresses most of my concerns. I am maintaining the current rating.

**Limitations:**

Yes

**Quality:**

2

**Strengths And Weaknesses:**

**Strengths**
- The work introduces a frequency‑aware token reduction strategy for ViTs, achieving state‑of‑the‑art results on multiple benchmarks.
- It offers a solid theoretical foundation in spectral analysis, linking the rank‑collapse phenomenon in transformers to the loss of high‑frequency components.
- The proposed method is computationally efficient and seamlessly integrates with existing ViT architectures, making it highly attractive for real‑world deployment.

**Weaknesses**
- **Clarity**: Some implementation details remain ambiguous—particularly the exact procedure for separating high‑ and low‑frequency tokens and the sequencing of global versus local operations in the token‑reduction pipeline. Including a more explicit algorithmic description or pseudo‑code would greatly enhance readability.
- **Visualization**: The work omits qualitative visualizations—such as token‑selection heatmaps or spatial maps before and after reduction—that are crucial for interpretability.
- **Typo**: There are some minor formulaic inaccuracies (e.g., $w^w$ should be $n/w²$), and some phrasing could be clearer regarding the frequency decomposition implementation.

---

> ### Author Rebuttal · Authors · 2025-07-31
>
> We sincerely thank the reviewer for their insightful summary and positive assessment of our work. We are grateful that you recognized our method's solid theoretical foundation, state-of-the-art results, and practical efficiency. Your valuable suggestions regarding clarity, visualization, and computational analysis are well-taken, and we will address each point to further strengthen the manuscript.
>
> - Clarity and Token Selection with Windowing
>
> We thank the reviewer for this suggestion. To clarify the procedure, our method first performs a global identification of High-Frequency (HF) and Low-Frequency (LF) tokens based on the full attention map, as described in Section 3.2 . Subsequently, the identified global LF token set is partitioned into spatial windows, and tokens within each windowed subset are aggregated into a *local* DC token . HF tokens are always preserved globally.
>
> To improve clarity and ensure reproducibility, we will add a dedicated algorithm block with pseudo-code in the Appendix of the revised paper.
>
> - **Visualization**
>
> Following your valuable suggestion, we performed an experiment to visualize the selected High-Frequency (HF) and Low-Frequency (LF) tokens on sample images. While we cannot attach figures directly to this rebuttal, we will describe our key findings here and commit to adding these visualizations in the main manuscript. Our analysis confirmed that in early to mid-layers, the token selection aligns well with human intuition: HF tokens predominantly correspond to patches with object boundaries and complex textures, while LF tokens are clustered in smooth, uniform background areas. Interestingly, we also observed 'outliers' where some background patches were selected as HF tokens, which we found often correlates with high-norm features in the embedding space. In deeper layers, we observed that where tokens corresponding to the salient foreground of an object began to be classified as LF tokens. Since rank collapse becomes more severe, the unique signals of even the most important tokens in human perception, get 'smoothed out' and converge toward the low-frequency, DC-like representation of the entire feature map.
>
> - **Role of DC Subtraction in Token Ranking**
>
> You are correct that the argmax ranking is mathematically unchanged by the subtraction. Our reasoning for this formulation is to maintain a clear and consistent theoretical framework throughout our method. This initial decomposition is crucial for consistency with our subsequent attention modification step in Equation (12), where we use a learnable parameter ω1 to amplify high-frequency signals . We will clarify this in the revised manuscript.
>
> - **Precise Definition of “Relative Magnitude”**
>
> The term “relative magnitude (weight)” in Eq.7 is calculated as the column-wise sum of the high-pass attention map corresponding to a particular token, which is then averaged across all attention heads. We will revise the text in Section 3.2 to make this connection explicit and avoid any confusion.
>
> - **Computational Cost Analysis**
>
> Thank you for this valuable suggestion. Following your recommendation, we have performed a more concrete computational cost analysis.
>
> A naïve method that involves calculating the DC signal and then measuring feature similarity between all tokens can have a complexity of up to $O(nd+nd^
> 2
> )$. In contrast, our proposed method, which simply post-processes the already computed attention map, requires only an additional $O(n^
> 2
> )$  complexity per head.
>
> For a typical model like DeiT-S with n=197 and d=384, the cost of our $O(n^
> 2
> )$  operation is significantly lower than that of the naïve approach. Even with large n, for instance, down stream task such as object detection, cost is still lower since $n << d^2$. This analysis confirms that our frequency-aware selection is not only highly effective, as demonstrated by our experiments, but also computationally very efficient. We will add this detailed cost analysis to the revised manuscript.
>
> - **Typo**
>
> Thank you for your careful reading and for pointing out the inaccuracies. We will correct all identified typos and unclear phrasing in the final version of the paper.

---

> > ### Comment · Reviewer_BU7E · 2025-08-05
> >
> > The response addresses most of my concerns. I am maintaining the current rating. Good work!

---

### Official Review · Reviewer_MPrk · 2025-06-20

**Clarity:** 3
**Significance:** 3
**Originality:** 3
**Rating:** 4
**Confidence:** 4

**Summary:**

To address the issue of quadratic computational complexity concerning token length in vision transformer, this paper proposed a frequency-aware token reduction strategy that improves computational efficiency while preserving performance by mitigating rank collapsing. Specifically, the proposed method partitions tokens into high-frequency (HF) tokens and low-frequency tokens (LF), which are selectively preserved and aggregated into a compact DC token to retain essential low-frequency components, respectively.

**Questions:**

Ablation on different image sizes: The paper mainly focuses on image-based token reduction, which may prove more effective when the sequence length is longer. Could the authors include ablation studies under different image sizes (e.g. 336$\times$336 or larger)?

**Ethical Concerns:**

["NO or VERY MINOR ethics concerns only"]

**Final Justification:**

The authors conducted addition experiments about generalization on the downstream task and ablation on different image sizes, and made a further explanation on HF/LF tokens, which addressed most of my previous concerns.

**Limitations:**

See `Weaknesses` and `Questions` part.

**Quality:**

3

**Strengths And Weaknesses:**

Strengths:
1. Well-motivated: The article presents a detailed discussion and visualization of the rank collapse or over-smoothing phenomena in self-attention and different token diversity of merging-based methods and reduction-based methods.
2. Clear writing: This paper offers a clear and logical structure that is easy to follow.

Weaknesses:
1. Limited performance improvement and downstream experiments: In `Tab.1` and `Tab.4` of the appendix, the authors compare their method with other approaches on classification and segmentation, respectively. However, the performance gain in `Tab.1` appears significant only on `ViT-S` and `ViT-S-21K`, while the improvements for other settings are relatively small. Furthermore, beyond the `baseline` comparison on the segmentation task, comparing with more closely related methods on additional vision downstream tasks (such as object detection) would better demonstrate the effectiveness of the proposed approach.
2. Visualization of HF/LF tokens: An important component of the proposed method is the division of tokens into high-frequency and low-frequency groups. Could the authors provide visualizations based on some images to offer a more intuitive interpretability analysis?


I'd be more than willing to revise my score based on the author's further responses.

---

> ### Author Rebuttal · Authors · 2025-07-31
>
> We sincerely thank the reviewer for the thorough and insightful feedback. Your suggestions have been invaluable in helping us strengthen our work. We address each of your points below with new experimental results and analyses.
>
> - **On Performance Improvement and Downstream Task Generalization**
>
> While the accuracy improvement is most visible on the ViT-S and ViT-S-21K models, we would like to clarify the goal of our method from a broader perspective, which demonstrates its effectiveness across all experiment settings.
>
> The primary objective of any token reduction method is to significantly reduce computational cost while minimizing performance degradation. From this viewpoint, the results for models like DeiT represent a major success. For instance, on DeiT-B, our method matches the baseline accuracy of 81.8% exactly, while reducing the computational cost from 17.6 to 11.6 MACs. This result is the ideal outcome, demonstrating that our core objective—maximizing efficiency without sacrificing performance—was successfully achieved. We consider this a significant contribution to the field of efficient models.
>
> The notable performance increase on ViT-S and ViT-S-21K can be seen as a significant benefit, highlighting a unique strength of our approach. As our paper analyzes, standard ViTs are prone to over-smoothing and rank collapse, where models lose token diversity.
>
> Following your valuable suggestion, we conducted additional experiments on object detection. To ensure a fair comparison, we adopted the experimental settings from SViT (Revisiting token pruning for object detection and instance segmentation) [1] and applied the same pruning configuration used in our main classification experiments (removing 30% of tokens at the 4th, 7th, and 10th layers). These experiments were performed on an A100 GPU. In the results table, values in bold represent our own measurements, while others are taken from the SViT paper for reference.
>
> The results show our method achieves a highly competitive performance-efficiency trade-off. Our method's APbox of 45.0 (-0.8 drop) is significantly better than other pruning-based methods like EViT (-1.3). When compared with SViT, which shows better performance preservation (-0.3 drop), it is important to consider the fundamental architectural differences that lead to this specific trade-off.
>
> SViT recycles information from pruned tokens in all subsequent layers, which helps preserve accuracy but introduces overhead. In contrast, our method is designed to be more lightweight by recycling this information only once at the final layer of the ViT-Adapter block. This key design choice is why our method achieves a notably higher throughput, running over 7% faster than SViT (21.0 vs 19.6 FPS), making it a compelling alternative for latency-sensitive applications.
>
> Furthermore, this trade-off is highly task-dependent. It is worth noting that on ImageNet classification with DeiT-S, our method outperforms SViT on both accuracy and efficiency. Our method achieves 79.8% accuracy (-0.0% drop), while SViT scores 79.4% (-0.4% drop).
>
> We hypothesize that adopting a more complex, SViT-style recycling mechanism could further improve our detection performance. While we were unable to complete this experiment due to time constraints, we aim to include this important analysis in the main manuscript of our final paper.
>
> | Method | APbox | APmask | FPS (Reported) | FPS (Measured) |
> | --- | --- | --- | --- | --- |
> | DeiT  | 45.8 | 40.9 | 18.45 | **18.02** |
> | SViT | 45.5 (-0.3) | 40.7 (-0.2) | 22.32 | **19.6** |
> | Ours | **44.9 (-0.9)** | **39.9 (-1.0)** | - | **21** |
> | EViT  | 44.5 (-1.3) | 39.8 (-1.1) | 22.76 |  | - |
> | EvoViT  | 44.8 (-1.0) | 39.9 (-1.0) | 22.12 | **19.3** |
> | DyViT + prsv | 44.1 (-1.7) | 39.3 (-1.6) | 22.95 | - |
>
> [1] Liu, Yifei, et al. "Revisiting token pruning for object detection and instance segmentation." *Proceedings of the IEEE/CVF Winter Conference on Applications of Computer Vision*. 2024.
>
> - **Visualization of HF/LF tokens**
>
> Following your valuable suggestion, we performed an experiment to visualize the selected High-Frequency (HF) and Low-Frequency (LF) tokens on sample images. While we cannot attach figures directly to this rebuttal, we will describe our key findings here and commit to adding these visualizations in the main manuscript. Our analysis confirmed that in early to mid-layers, the token selection aligns well with human intuition: HF tokens predominantly correspond to patches with object boundaries and complex textures, while LF tokens are clustered in smooth, uniform background areas. Interestingly, we also observed 'outliers' where some background patches were selected as HF tokens, which we found often correlates with high-norm features in the embedding space. In deeper layers, we observed that where tokens corresponding to the salient foreground of an object began to be classified as LF tokens. Since rank collapse becomes more severe, the unique signals of even the most important tokens in human perception, get 'smoothed out' and converge toward the low-frequency, DC-like representation of the entire feature map.
>
> - **Ablation on Different Image Size**
>
> Thank you for this excellent suggestion. Following your advice, we have conducted an extensive ablation study on various image and patch sizes to analyze our method's effectiveness on different token lengths. For these experiments, we used the ViT-21k-finetuned-on-1k model, as it provides pre-trained weights across the various configurations. The results, summarized in the table below, strongly support our hypothesis and demonstrate the remarkable scalability of our approach. (GFLOPS are shown in parentheses).
>
> | patch size / image size | 32/224 | 32/384 | 16 / 224 | 16 / 384 | 8/224 |
> | --- | --- | --- | --- | --- | --- |
> | image token length  | 49 | 144 | 196 | 576 | 784 |
> | ViT-B (original) | 80.7 (4.4) | 83.4 (13.1) | 84.5 (17.6) | 86.6 (55.5) | 85.8 (78.3) |
> | ViT-B (pruned) | 77.9 (3.2) | 84.0 (8.8) | 84.6 (12.1) | 87.0 (35.9) | 86.5 (50.0) |
> | Difference | -2.8 (-1.2) | 0.63 (-4.3) | 0.14 (5.5) | 0.4 (-19.6) | 0.7 (-28.3) |
> |  |  |  |  |  |  |
> | ViT-S (original) | 76.0 (1.1) | 80.5 (3.4) | 81.1 (4.6) | 83.8 (15.5) |  |
> | ViT-S (pruned) | 73.5 (0.8) | 80.5 (2.3) | 81.4 (3.0) | 84.0 (9.9) |  |
> | Difference | -2.5 (-0.3) | 0.0 (-1.1) | 0.30 (-1.6) | 0.16 (-5.5) |  |
>
> As you hypothesized, the benefits of our method are most prominent with longer token sequences. The results clearly show that our approach not only improves efficiency but can also enhance performance, especially when dealing with high token counts.
>
> For the high-resolution case (16/384, 576 tokens), our method reduces the computational cost by a substantial 35% (from 55.5 to 35.9 GFLOPS) while also improving the accuracy from 86.6% to 87.0%.This trend is even more pronounced with the longest sequence tested (8/224, 784 tokens), where we achieve a +0.7% accuracy gain while cutting GFLOPS by 36% (from 78.3 to 50.0 GFLOPS).
>
> We thank you for pushing us in this valuable direction. These new experiments clearly demonstrate the effectiveness and scalability of our method. We will add this full ablation study to the final version of our paper.

---

> > ### Author Response · Authors · 2025-08-04
> > **Follow-up comments to Reviewer MPrk**
> >
> > Dear Reviewer MPrk,
> >
> > Thank you again for your thorough and insightful review. We truly appreciate the time and expertise you've dedicated to our manuscript.
> >
> > We have submitted our detailed rebuttal, which includes several new experiments specifically designed to address the crucial points you raised regarding the performance improvement, generalization on the downstream task, visualization, and ablation on different image sizes. We hope our response and the new results have clarified these aspects of our work.
> >
> > We are eager to know if our response has adequately addressed your concerns. If any points remain unclear or if further clarifications are needed, we would be grateful for the opportunity to discuss them. If additional experiments are needed, we're happy to run them, though please note that due to compute time, we may not be able to deliver results before the discussion window closes.
> >
> > Thank you once more for your valuable feedback.
> >
> > Sincerely,
> >
> > The Authors

---

> > ### Comment · Reviewer_MPrk · 2025-08-05
> >
> > Thanks for authors' detailed response and additional experiments, which addressed most of my previous concerns. So I decide to raise my score to 4.

---

### Official Review · Reviewer_pKDi · 2025-06-30

**Clarity:** 2
**Significance:** 2
**Originality:** 3
**Rating:** 4
**Confidence:** 4

**Summary:**

This paper proposed a token reduction method based on frequency analysis. The low-pass filter by a mean filter is applied to attention map. Then, the high-frequency attention map is obtained by subtraction between the original attention map and low-pass filtered map. Then, reduced tokens are selected based on the highest weights in the HF attention map. To maintain the important information from low-frequency tokens, DC signals of LF tokens are aggregated into a single DC token. The experiments are conducted for pre-trained models trained on ImageNet-1K.

**Questions:**

1. Is it enough to use the mean filtering to obtain the low-pass filtered output of attention maps?
2. Why did you compare your results with somewhat less recent methods?
3. Are there any results applied to datasets other than ImageNet?
4. Is there any reason to limit 30 epochs in the experiments?

**Ethical Concerns:**

["NO or VERY MINOR ethics concerns only"]

**Final Justification:**

I understand why you did not compare it to other 'merging' or 'pruning' techniques. Furthermore, additional experiments alleviated my previous concerns. I will raise the score to 4.

**Limitations:**

Yes.

**Paper Formatting Concerns:**

No.

**Quality:**

2

**Strengths And Weaknesses:**

Strengths
1. The idea of using frequency information when selecting tokens is effective.
2. The computational overhead is not severe to select high frequency tokens.
3. Better than previous methods for various models of DeiT, ViT.

Weaknesses
1. As in Eq. (5), the mean filtering of attention maps is not guaranteed to produce effective low-pass filtered output. Thus, it is not easy to have successful high-pass filtered output. We cannot expect that this simple filtered outputs help to select informative tokens. Similarly, in the process of aggregating DC signals of LF tokens, it is not clear that we can have DC signals by repeatedly aggregating the LF tokens from the mean filtering.
2. Most of previous methods in Table 1 are published 2 or 3 years ago. Therefore, it is not easy to evaluate that the proposed method outperform the previous state of the art methods. For ViTs, the previous methods are ToMe and EViT, which are published in 2022. There are more recent methods for token reduction, PYRA(ECCV'24), DyT(Neurips'24), OFB (CVPR'24), etc.
3. It is difficult to determine whether this method can be applied generally because it is only applied to one dataset, ImageNet. It would be better to apply the proposed method to other vision problems, such as VTAB-1K or CIFAR10, Cityscape.
4. Usually, we need to train at least 100 epochs in previous token reduction methods. 30 epochs may not be enough to compare the performance. There is no clear explanation why 30 epochs are used for the experiments.

---

> ### Author Rebuttal · Authors · 2025-07-31
>
> We thank the reviewer for the feedback. Below, we aim to address you concerns regarding our methodology and experimental validation to further clarify the contributions of our work.
>
> - **On Experimental Setup, Baselines, and SOTA Comparisons**
>
> We would like to address the concerns regarding the suggested recent works, datasets, and training epochs.
>
> First, we thank you for suggesting the recent papers. However, we would like to clarify that the mentioned works—PYRA and DyT focusing on 'Efficient Task Adaptation,' and OFB on 'Model Compression'—belong to research fields that are orthogonal to our work on 'token reduction for pre-trained models'. These methods aim to improve the efficiency of 'parameter-efficient fine-tuning' or 'compression' via adaptor or network sparsity search, whereas our work concentrates on dynamically reducing the number of input tokens during 'inference'. In fact, the DyT paper itself suggests this complementary relationship by stating, "Then, we explore the compatibility of our method with token pruning methods," indicating that our fields are not direct comparable methods.
>
> As these research fields differ, so do their standard evaluation protocols. As the reviewer noted, papers in those fields often evaluate on the VTAB-1K dataset with 100 training epochs. However, the standard for our subfield—as followed by recetn SOTA methods (e.g., DTEM('24), Zero-Tp('24)) in Table 1 —is 30-epoch fine-tuning on ImageNet-1K. We adhered to this protocol to ensure a fair and direct comparison. Furthermore, we chose DeiT as our primary comparison model among recent SOTA methods, because it is the standard model used in recent token reduction literature. Therefore, we compared the proposed method with recent SOTA methods, in DeiT. The inclusion of the ViT model was not reported in the most of the previous works, but was a necessary choice to validate our core hypothesis and proposed method; that the degree of rank collapse differs depending on the training recipe, thus requiring verification under different conditions. In addition, the practical challenge of some recent works not having publicly available code also factored into our decision.
>
> In conclusion, our reason for selecting ToMe / DynamicViT, EViT as primary baselines is that they represent the two foundational pillars of token reduction: 'merging' and 'pruning'. While most subsequent works are incremental improvements on their ideas, our research introduces a fundamental difference by approaching the problem from a new aspect based on 'frequency'. Our goal was to extensively validate the efficacy of this novel methodology across various models, and we respectfully submit that our experimental design was therefore reasonable and well-justified. We hope this explanation helps to address the reviewer's concerns.
>
> Nevertheless, to further address the reviewer's concern and further demonstrate the effectiveness of our method, we have conducted new experiments with DyT and AdaptFormer. To ensure fairness, we replicated the experimental environment from Table 5 of the DyT paper (using AdaptFormer on VTAB-1K). Since the DyT paper lacks a precise description of its token pruning setup for that table, we followed the original ToMe experiment settings by pruning 13 tokens in each block. Values measured by us are marked in **bold**; otherwise, they are from Table 5 of the DyT paper. VTAB Accuracy refers to the grouped-mean accuracy, and the parameter count denotes the number of trainable parameters in the backbone.
>
> | Method | VTAB-1K Accuracy ↑ | Params. (M) ↓ | Throughput (img/s) (V100) | Throughput (img/s) (A100) |
> | --- | --- | --- | --- | --- |
> | AdaptFormer | 74.75 | 0.16 | 767.3 | **3296** |
> | DynamicViT+AdaptFormer[12] | 75.48 | 3.1 | 954.82 |  |
> | EViT+AdaptFormer[12] | 74.63 | 0.16 | 1152.38 |  |
> | ToMe + AdaptFormer | 74.3 | 0.16 | 941.7 |  |
> | Ours + AdaptFormer | **74.71** | **0.16** |  | **4752** |
> |  |  |  |  |  |
> | DyT r= 0.5 | 77.14 | 0.16 | 912.39 | **3478** |
> | DyT r= 0.5 + ToMe [5] | 76.6 | 0.16 | 1114.7 |  |
> | DyT r= 0.5 + Ours | **76.77** | **0.16** |  | **4859** |
>
> The result clearly shows that our frequency-aware approach is a effective solution that surpasses existing token reduction methods, even when integrated with the latest SOTA adaptation frameworks.
> These experiments on VTAB-1K, together with our results on ImageNet-1K and the ADE20K segmentation task (in supplementary materials), clearly demonstrate the robustness and effectiveness of the proposed method across diverse datasets and tasks.
>
> - **On the Validity of Mean Filtering**
>
> Our use of mean filtering ($A^{LP}$) to isolate low-frequency components is a direct application of a fundamental signal processing principle: a signal's mean is its DC (zero-frequency) component, a quintessential low-pass operation (see Appendix C for a formal proof ). In addition, the mean filter is widely used for a lowpass frequency filter, effectively removing high-frequency content from an image, or token, in our papers [1,2].
>
> While our selection method is computationally simple, its effectiveness is empirically validated in Figure 2, which confirms that our selected high-frequency (HF) tokens are indeed richer in high-frequency content and more critical to model accuracy. Furthermore, its effectiveness is validated by the high fidelity of our aggregated DC token to the original DC signal (Figure 4) and the notable performance drop upon its removal (Figure 5). We believe this strong theoretical and empirical evidence demonstrates the soundness of our approach.
>
> [1] Stanford University. (n.d.). CS279 Notes.
>
> [2] Fisher, R., Perkins, S., Walker, A., & Wolfart, E. (n.d.). Hypermedia Image Processing Reference (HIPR2). University of Edinburgh.

---

### Official Review · Reviewer_PN7n · 2025-07-03

**Clarity:** 3
**Significance:** 4
**Originality:** 4
**Rating:** 5
**Confidence:** 5

**Summary:**

The manuscripts describes a novel token reduction method based on prior research on overshooting and rank collapse of self-attention / transformer models, and additional analysis of these phenomena as token reduction tends to exasperate these conditions. In order to avoid this a frequency-aware token reduction method is proposed, where the input tokens are partitioned into two subsets: a low frequency (LF) set of tokens, which are similar to the DC Bias (mean token value), and high frequency (HF) set consisting of tokens which are dissimilar to the DC bias. Furthermore, chunked attention is initially used in order to account for spatial effects, as the LF tokens are found to contain HF signals in early layers.

The methods is trained using a teacher-student distillation setup and compared across several pretrained backbones (ViT, DeiT, MAE, DINO) on ImageNet-1K. The proposed method is found to consistently matching and outperforming prior methods. Lastly, analysis and ablation studies of the effect and behavior of the LF and HF tokens are conducted, and prior token reduction methods are analyzed in a frequency perspective.

**Questions:**

1. Why are the results on dense prediction tasks (section G in the supplementary results) not described more in the main paper? The results, as with the other sections, are interesting and currently the reader may not fully recognize this when reading the main manuscript.

2. Does the reduction rate r have to be the same for both LF and HF subsets? Have you conducted any investigation into this, and whether it could be made dynamic such that all relevant tokens could be used?

3. What is the effect of the distillation based training vs standard fine-tuning? Clarifying this would make the results more certain, as currently the effect of the distillation setup and the frequency-aware token reduction is entangled.

**Ethical Concerns:**

["NO or VERY MINOR ethics concerns only"]

**Final Justification:**

Having read the author's responses to my own and the other reviews I have found all of my concerns to be answered. i believe the paper is of high interest to the Token Reduction field and provide interesting theoretical and empirical results. I am therefore raising my recommendation to Accept.

**Limitations:**

yes

**Paper Formatting Concerns:**

1. In Eq 7 it is not explicitly stated that h refers to the attention head index.

2. Line 138-140 becomes a bit convoluted and could benefit from rewriting and simplification.

3. In line 469 the closing parentheses are missing.

**Quality:**

4

**Strengths And Weaknesses:**

Strengths:

1. The proposed method is in essence very simple, as it simply consists of disentangling the LF and HF parts of the attention maps. This makes the method easy to adopt and understand.

2. The method is founded on theoretical and empirical results related to the rank collapse and oversmoothing of self-attention layers. This again leads to the method to be easily understood, and also serves as a tool for interpreting prior methods as done in Section 6.3.

3. As shown in the experiment section, the proposed method matches or in some cases significantly outperforms prior token reduction methods. This is impressive, considering the large suite of methods compared against.


Weaknesses:

1. It is unclear why a student-teacher distillation setup is used. It is not motivated by any part of the manuscript and the addition leads to uncertainty of whether the results are due to the proposed frequency-aware method or the distillation training setup.

---

> ### Author Rebuttal · Authors · 2025-07-31
>
> We sincerely thank the reviewer for their thorough feedback and insightful comments. We are encouraged that they found our method to be simple, theoretically sound, and experimentally effective. Below, we address each point raised and believe these clarifications will help strengthen the final manuscript.
>
> - **Effects of self-distillation**
>
> We adopted self-distillation to follow the standard protocol for a fair comparison. The main prior works we compare against, such as EViT ,Evo-ViT , and DynamicViT, also report their performance using this same self-distillation method. This approach is standard practice in the field.
>
> To  address the concern, we ran an additional experiment without any distillation. The results confirm its impact is minimal: the performance DeiT-S w/ distillation (79.8%) and w/o distillation (79.7%). The similar behavior can be observed in the other methods, for instance, DynamicViT w/ distillation (79.3%) and w/o distillation (79.2%).
>
> This ablation study demonstrates that the performance gains stem from our proposed frequency-aware method, not the distillation setup. We will add this analysis to the final paper and thank you for helping us strengthen our work.
>
> - **Reduction rates of LF and HF tokens.**
>
> We use a fixed reduction rate, to precisely control the computational budget (MACs). This is a standard practice in the field that ensures a fair comparison with prior work. Our method then preserves the top-r high-frequency (HF) tokens.
>
> Our paper's primary contribution is the novel frequency-aware selection criterion, which is orthogonal to the scheduling method (fixed vs. dynamic) . While we are aware of dynamic methods like A-VIT and ATS , our focus was on first validating this new criterion. We agree that a dynamic approach is a promising direction. However, dynamic methods introduce implementation challenges, particularly for batched training and inference, as token lengths can vary per sample. We are actively exploring how to efficiently integrate our criterion with a dynamic scheduler, and if the experiments yield meaningful results in time, we will include them in the final manuscript. This investigation builds upon our initial exploration of optimal reduction schedules via NAS (Appendix F) .
>
> - **Dense Prediction tasks**
>
> We thank the reviewer for this excellent suggestion. The semantic segmentation results were moved to Appendix G primarily due to the main paper's strict page limits. We prioritized a detailed analysis of our core methodology on the primary benchmark of image classification.
>
> However, we agree that these results are an important demonstration of our method's generality. In the revised manuscript, we will add a summary of the dense prediction results to the main paper.
>
> - **Paper Formatting**
>
> Thank you for your review and for catching these details. We appreciate you helping us improve the clarity of our paper. We will correct all the formatting issues you pointed out in the final version.

---

> > ### Comment · Reviewer_PN7n · 2025-08-04
> > **Response to authors**
> >
> > Dear authors,
> >
> > I have read your response, as well as the responses to the author reviewers, and find that my concerns have been responded to in a satisfactory manner. I am therefore raising my rating to Accept.

---

### Note · Authors · 2025-08-15

We sincerely thank the Area Chair and all reviewers for their insightful and constructive feedback. The review process has been incredibly valuable in helping us strengthen our work.

We are delighted that three reviewers (PN7n, MPrk, and BU7E) found our rebuttal and extensive new experiments convincing, leading them to raise or maintain their positive ratings. We believe the discussion period has allowed us to substantially improve our paper and further validate the effectiveness of our proposed method.

During the discussion period, the three participating reviewers (PN7n, MPrk, BU7E) primarily raised questions regarding the impact of self-distillation, generalization to downstream tasks like object detection, scalability with longer token sequences, and points of clarification on our methodology. We addressed these comprehensively with extensive new experiments, including an ablation on self-distillation, performance validation on the COCO object detection benchmark, and a large-scale analysis across various image and patch sizes to demonstrate scalability. We are sincerely grateful to the reviewers for their insightful feedback, which has been instrumental in enhancing the quality of our manuscript.

All three reviewers confirmed their concerns were fully resolved, with two (PN7n, MPrk) raising their scores and one (BU7E) maintaining their positive rating. While Reviewer pKDi's final comment was unavailable, we provided a detailed rebuttal clarifying our field's standard evaluation protocols and included new experiments on the VTAB-1K dataset to directly address their concerns. We are confident these additions adequately resolve the points raised in their initial review.

We are confident that these new results and clarifications have thoroughly addressed the key concerns raised by all reviewers. We believe our work presents a novel, theoretically-grounded, and empirically-validated contribution to efficient vision transformers. We look forward to incorporating this feedback to produce an even stronger final paper. Thank you for your consideration.

---

### Decision · Program_Chairs · 2025-09-17

**Decision:**

Accept (poster)

**Comment:**

The paper proposes a strategy to alleviate the high (quadratic) computational cost of vision transformers by performing token reduction.
Importantly, the token reduction is done in a frequency-aware way to avoid the common side effects: rank collapsing and over-smoothing: The tokens are divided into two categories: High-frequency tokens and low-frequency tokens (corresponding to direct current (DC)/bias/zero frequency), which are then handled differently.
Through extensive experiments, the paper shows that the proposed method outperforms existing baselines.

During rebuttal and discussion, the authors provided additional details about the procedure and experiments, which greatly helped convince the reviewers.
Overall, the reviewers generally liked the paper for its simplicity, theoretical grounding, and extensive experiments that demonstrated practical benfits.
For the camera-ready, the paper should incorporate the clarifications and improvements discussed during review.